# Exploring the relation between modelled and perceived workload of nurses and related job demands, job resources and personal resources; a longitudinal study

Wilhelmina F. J. M. van den Oetelaar [1]*, Corné A. M. Roelen[2], Wilko Grolman[3], Rebecca K. Stellato[4], Willem van Rhenen[2,5]

1 Division of Internal Medicine, University Medical Center Utrecht, University of Utrecht, Utrecht, the Netherlands, 2 ArboUnie Occupational Health Service, Utrecht, the Netherlands, 3 Division of Surgical Specialties, University Medical Center Utrecht, University of Utrecht, Utrecht, the Netherlands, 4 Julius Center for Health Sciences and Primary Care, University Medical Center Utrecht, University of Utrecht, Utrecht, the Netherlands, 5 Center for Human Resource Organization and Management Effectiveness, Business University Nyenrode, Breukelen, the Netherlands

* W.f.j.m.vandenoetelaar@umcutrecht.nl

**Data Availability Statement:** All relevant data are within the manuscript. Data are also available in the OSF public data repository via permalink https://

# Abstract

## Aim

Calculating a modelled workload based on objective measures. Exploring the relation between this modelled workload and workload as perceived by nurses, including the effects of specific job demands, job resources and personal resources on the relation.

## Design

Academic hospital in the Netherlands. Six surgical wards, capacity 15–30 beds. Data collected over 15 consecutive day shifts.

## Methods

Modelled workload is calculated as a ratio of required care time, based on patient characteristics, baseline care time and time for non-patient related activities, and allocated care time, based on the amount of available nurses. Both required and allocated care time are corrected for nurse proficiency. Five dimensions of perceived workload were determined by questionnaires. Both the modelled and the perceived workloads were measured on a daily basis. Linear mixed effects models study the longitudinal relation between this modelled and workload as perceived by nurses and the effects of personal resources, job resources and job demands. ANOVA and post-hoc tests were used to identify differences in modelled workload between wards.

## Results

Modelled workload varies roughly between 70 and 170%. Significant differences in modelled workload between wards were found but confidence intervals were wide. Modelled workload is positively associated with all five perceived workload measures (work pace,

osf.io/hvp9j/?view_only=
f9d51897bcf54540941e69b53ba1eb29.

**Funding:** The authors received no specific funding for this work.

**Competing interests:** The authors have declared that no competing interests exist.

amount of work, mental load, emotional load, physical load). In addition to modelled workload, the job resource support of colleagues and job demands time spent on direct patient care and time spent on registration had the biggest significant effects on perceived workload.

## Conclusions

The modelled workload does not exactly predict perceived workload, however there is a correlation between the two. The modelled workload can be used to detect differences in workload between wards, which may be useful in distributing workload more evenly in order prevent issues of over- and understaffing and organizational justice. Extra effort to promote team work is likely to have a positive effect on perceived workload. Nurse management can stimulate team cohesion, especially when workload is high. Registered nurses perceive a higher workload than other nurses. When the proportion of direct patient care in a workday is higher, the perceived workload is also higher. Further research is recommended. The findings of this research can help nursing management in allocating resources and directing their attention to the most relevant factors for balancing workload.

## Introduction

In healthcare, there is an increasing pressure on maintaining a high quality of care whilst containing expenditure [1]. Healthcare expenditure is increasing due to factors such as technological progress and an aging population with more chronic conditions [2, 3]. Healthcare providers are expected to maintain a high quality of care under increasing demand, with the same or less funding [4–6]. There is a risk that under these circumstances, nurses' workload will increase to alarming levels. Workload is considered unacceptable, when nurses are not able to meet patients' needs, physical as well as emotional [7], or if nurses' health is at risk. There is overwhelming evidence of the adverse effects of high workload of nurses. There is a direct relation between nurses' workload and patient satisfaction [8], patient outcomes [9–14] and nurse reported quality of care and performance [15–17]. High workload is also a predictor for nurses' job dissatisfaction, stress, burnout [18–20] and absenteeism [21], and generally shows a negative effect on job outcomes [22] and nurses' intention to leave [23, 24]. High turnover of nursing staff results in higher costs for training of new nurses, or hiring of temporary staff [11, 25]. Nurses also report that when workload is high, they cannot deliver all the care that they believe they should [26]. Important tasks, such as tending to patients' emotional and psychological needs, are left undone, which leaves nurses feeling dissatisfied with their job and occupation,. In turn, this leads to an increasing intent to leave, lower reported quality of care and deteriorating patient satisfaction [26].

Managing workload will therefore become increasingly important in preventing the vicious cycle where nurses are leaving the job due to high workload, which leads to a greater shortage and eventually an even higher workload.

Despite the fact that a high workload can have such far-reaching practical implications, the concept of workload was not always clearly defined in research. It was often measured by using crude staffing ratios or determined by questionnaires regarding perceived workload. Also, many studies examined only one dimension of workload, such as mental load or amount of work [27]. Alghamdi [28] defined workload as the amount of time and care that a nurse can

devote (directly and indirectly) towards patients, workplace and professional development. This definition covers direct and indirect patient care but also non-nursing activities such as meetings and attending seminars. Alghamdi advocated a holistic approach and determined five defining attributes of nursing workload: the amount of nursing time spent on nursing care known as patient acuity; the level of nursing competency; the weight of nursing intensity (direct patient care); all the physical, mental and emotional efforts; and the ability of the nurse to change the plan (complexity of care). Holden [29] described three different types of perceived workload: task-level, job-level and unit-level workload. Unit-level workload consider the balance between patient acuity and staffing, job-level workload entails general and specific demands of the job such as the general amount and difficulty of the work and the amount of concentration required to do it, and task-level workload relates to the demands and resources for a specific task such as medication preparation. These workload types describe different dimensions of workload and each type of workload has a specific effect on burnout, job dissatisfaction and the likelihood of medication errors. Holden's study did not consider emotional and physical workload, but recommended also taking these into account. In the Netherlands, the Questionnaire on the Experience and Evaluation of Work (QEEW [30]), an evidence based questionnaire, is widely used by Dutch occupational health services to measure the (psychosocial) working environment. It defines four types of workload: work pressure (a combination of work pace and amount of work), emotional load, mental load and physical effort.

Previous research of our study group [31] suggested a method to model nurses' workload, based on objective measures such as patient characteristics. In the study presented in this paper, we have built on the findings of this previous research. Both objective (calculated, modelled) and subjective (perceived, self-reported) workload measures were included in the study.

We were interested to see whether the modelled workload correlates with perceived workload of nurses. The perceived workload measures from the QEEW were included in our study. The measures are well known in the Netherlands and also emotional and physical components of workload were included, which are important aspects of nursing. We have studied the two components of work pressure (work pace and amount of work) separately though. The modelled workload may have a different relation with the two perceived workload measures. For example if the amount of work is perceived to be high, this may not automatically result in a higher work pace, since tasks may be left undone.

In addition to considering the relation between modelled and perceived workload, we were also interested to research the effects of personal factors and work environment on this relation.

The well-known Job Demands and Resources (JDR) model [32] also considers workload and we have used this model as a framework for our study. The JDR model describes a health impairment process where high job demands lead to exhaustion and burnout. Job demands are those aspects of a job that require effort. On the other hand, the JDR model describes a motivational process where job resources play a crucial role. Job resources are functional in achieving work goals, reduce job demands and the associated physiological and psychological costs and stimulate personal growth and development [32]. Workload is considered a job demand in the JDR model; prolonged periods of high workload can lead to burnout, especially if there are insufficient job resources to counterbalance the effects of the job demands [32–34].

The aim of our study was to understand the relation between objective, modelled workload and workload as it is perceived by nurses, and to test the effects of several demands and resources that are considered relevant in this context, according to the available literature.

This paper considers the effects of four job demands, three job resources and two personal resources on the relation between modelled and perceived workload.

Perceived interruptions, perceived equal work distribution, time spent on registration and time spent on direct patient care are considered relevant job demands in the context of this study. Dutch occupational health surveys, used by occupational health and safety services for preventive medicine purposes, often include questions on work interruptions and equal work distribution. There is evidence that strongly relates the number of work interruptions to patient outcomes [35, 36] and nurses' workload [37, 38]. Equal work distribution has not been related to perceived workload in literature before. Registration required by the government, insurance companies and hospital management, is one of the reasons nurses experience a high workload. Myny [38] found a link between perceived workload and the number of mandatory registrations. Van Bogaert [7] also identifies a growing problem of additional registration. There are also indications that increased administration burden competes with the time intended to be spent on patient care; Khademi [39] stated that one of the important sources of increased nursing workload was an overemphasis of managers on frequent report writing, which competed with patient care delivery. Workload may also be experienced differently if the proportion of time that nurses can spend on direct patient care is low, due to either increased registration or other reasons.

The JDR model postulates that the health impairment process can be mitigated by job resources and personal resources [32]. These resources help to achieve goals and stimulate personal development, resulting in intrinsic motivation and engagement [32, 34, 40–43].

The present study has included three job resources: support from colleagues, support from management, and the proportion of registered nurses on the ward. Yanchus [44] found that the job resource teamwork (i.e, colleagues helping and backing each other up) counterbalances the effects of understaffing and high workload. Similarly, Sexton's research [45] found that workload pressures can be offset by a positive nursing team environment on a unit. This indicates that teamwork is an important factor in perceived workload. Van Bogaert et al. [7] found the same results in their study on the predictors of burnout, work engagement, nurse reported job outcomes and quality of care. Van Bogaert et.al. [46] also reported that hospital management directly influenced nurses' perceived workload. In MacPhee's study on the impact of heavy perceived workload of nurses on patient and nurse outcomes [14], it was noted that unit level leaders in particular could influence perceived quality of care and job outcomes by monitoring and responding to workload demands. In addition, there is much evidence that a nursing staff mix with a large proportion of registered nurses results in better patient outcomes [36, 47, 48]. The effect of the skill mix of nurses on perceived workload has not been extensively studied, although there is evidence that skill mix is an important factor when considering workload [49] and that a lower proportion of registered nurses on a ward is associated with increased workload [50]. In the current paper, the proportion of registered nurses on the ward was regarded as a job resource and was included in the study.

The research also included two personal resources: self-efficacy and proficiency. Spence Laschinger [51] found a significant correlation between workload (as one of the areas of work life) and occupational coping self-efficacy, defined as the self-appraisal of one's capability to cope with occupational burden in the workplace [52]. When the areas of work life, such as workload, are balanced, this has a positive effect on occupational coping self-efficacy. In the hospice setting, the stress of staff shortages decreased with increasing self-efficacy [53]. Schmidt et al. [41] found that self-efficacy had a direct effect on job strain in nursing homes, but they did not find evidence for an interaction between self-efficacy and perceived workload. Brunetto [54] found that self-efficacy reduced the effects of stress and enhanced job satisfaction, and herewith reduced nurses' intentions to leave the job; however, they did not study the effects on perceived workload.

Nurses' education and experience are related to clinical expertise [55]. A well-educated, experienced nurse may be able to handle workload better than a novice nurse. We have tested whether nurses with more working experience and who are considered to be more proficient experience workload differently than other nurses.

We have tested the following hypotheses:

• *Hypothesis 1*: there is a positive correlation between the modelled (objective) workload measure and perceived (subjective) workload

• *Hypothesis 2*: perceived workload is lower when personal resources self-efficacy and nurse proficiency are higher.

• *Hypothesis 3*: the job resources support from management, support from colleagues and proportion of registered nurses moderate the relation between modelled and perceived workload.

• *Hypothesis 4*: the job demands proportion of direct patient care, proportion of administration, work interruptions and perceived equality of work distribution moderate the relation between modelled and perceived workload.

We expected that the job resources and job demands mentioned under hypotheses 3 and 4 diminish the effect of increasing objective workload on the perceived workload. They may act as a buffer and therefore we have tested moderation.

## Materials and methods

### Study setting

The research took place in an academic hospital in the Netherlands (private hospital). Six surgical wards were included (2 wards with 15 beds, 4 wards with 30 beds). The study focused on the day shift workload, because this is the shift during which the most nurses are required and most nursing activities are performed, some of which only occur in the dayshift.

Weekends were excluded because the task mix and staffing are very different in weekends and cannot be compared to the day shifts of regular weekdays.

### Participants

All registered nurses and nurse students working on the study wards were included in the study. Ward managers were excluded because they do not perform patient-related activities. The study focused on workload of nurses: other professionals such as physicians, physician-assistants, and paramedics were not included in this study.

Factors that consider employees other than nurses (such as nurse-physician relationships, support from logistic teams), factors that cannot be influenced by nurse management of the ward (such as social support at home) or factors that require major investments (such as ward layout and number of single rooms in a ward) were not included.

### Design

In this research, five different measures of perceived workload were included: work pace, amount of work, mental workload, emotional workload and physical workload. The job demands (interruptions, work distribution, time for registration, time for direct patient care) and job resources (support from colleagues, support from management, the proportion of registered nurses on the ward) were included in the analyses as potential effect modifiers of the

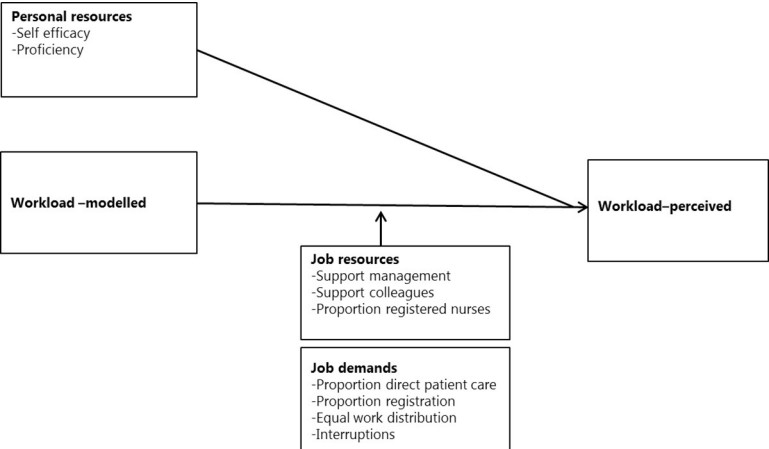

**Fig 1. Studying the relation between modelled and perceived workload (our hypotheses).**

relation between modelled and perceived workload. Effects were explored separately for each outcome measure. **Fig 1** presents a visualization of the model.

## Data collection

Data was collected in four ways:

1. An observational work sampling study of the activities of nurses was done during fifteen consecutive day shifts. Details on the study can be found in a previous publication of the study team [56]. This study yielded estimates of time spent on direct patient care, collective patient care, general tasks and other tasks.

2. During the observational study period, a previously determined set of relevant patient characteristics were registered each dayshift for all patients admitted on the study wards [57]. This information was combined with the work sampling results and a linear mixed effects model was used to determine significance and care time estimates for each characteristic [57, 58].

3. During the observational study, each nurse on duty during the dayshift was asked to fill out a short questionnaire at the end of each dayshift. The questionnaire was printed and handed out by the lead researcher and contained questions on job demands, job resources, engagement, quality of care and stress.

4. Three weeks prior to the observational study, all nurses employed on the study wards (as registered nurses or students) were asked to fill out an extensive online questionnaire (via Survey Monkey®) on job demands, job resources, personal resources and personal and job characteristics.

## Measures

**Outcome measures: Perceived workload.**   Perceived workload was measured with items derived from the Questionnaire on the Experience and Evaluation of Work (QEEW), which is widely used by Dutch occupational health services to measure the (psychosocial) working environment [41]. The five measures for perceived workload were:

'Did you have too much work to do today', reflecting the amount of work;

'Did you have to work very fast today', reflecting work pace;

'Did you consider your work mentally very challenging today', reflecting mental workload;

'Did your work demand a lot from you emotionally today', reflecting emotional workload;

'Did you find your work physically strenuous today', reflecting physical workload.

Each item had a five-point response scale ranging from 'Not at all' (1) to 'Very often' (5). All measurements were done during the work sampling study period, on a daily basis at the end of the shift.

**Independent variable: Modelled workload.** Previous work of this study team described a study design for developing a workload management method [31]. This method consists of 7 steps to calculate workload. Steps 1 to 4 were elaborated upon in previous publications [31, 56, 57] and in this paragraph we will briefly explain these steps. Steps 5 to 7 will be discussed in depth in this paper.

In step 1, a group of expert nurses composed a list of fifteen patient characteristics that they expected to be most relevant to care time. Subsequently (step 2), an observational study on nurses' activities measured how much time nurses spent on direct patient care, collective patient care, general tasks and other tasks [56]. The relation between the patient characteristics and observed care time was analyzed in step 3 [57]. Nine patient characteristics significantly increased care time. This study also showed that it was important to incorporate nurse proficiency in care time estimates and workload calculations. For this reason, observed care times were corrected for nurse proficiency in the source data of the work sampling study. Proficiency was estimated by expert nurses. In a mini Delphi study, all six ward managers of the wards involved in the study were asked to define nurse categories and corresponding proficiency levels. The mini Delphi consisted of two rounds and two sessions to discuss results. The Delphi yielded six nurse proficiency categories. The reference category is the fully qualified and experienced registered nurse: this was defined as the standard and set to a proficiency percentage of 100%. The proficiencies of the other five types of nurses (novice registered nurses, fulltime student nurses in their first or second year, fulltime student nurses in their third or fourth year, working student nurses in their first or second year, working student nurses in their third or fourth year) were offset against this standard.

In addition to care time related to patient characteristics, it was also assumed that there is a patient-related 'baseline care time'. When a patient is admitted to a ward, nurses will always spend a certain amount of care time on this patient, regardless of the patient characteristics that apply. For example time that is spent on handing out meals, having a chat or tidying up. The baseline care time was also estimated in step 3. The observational study also yielded how much time nurses spent on non-patient related activities (step 4).

In this workload management method, the modelled workload (step 7) was calculated by comparing required nursing time (step 5) to allocated nursing time (step 6). The total required nursing time was the result of adding up care time related to patient characteristics, baseline patient related care time and time for non-patient related activities:

:

$$(a1 * n1 + a2 * n2 + a.. * n.. + ..) + (b * N) = x$$

$$Yreq = \frac{x}{(1-z)}$$

$$a1 = additional\ care\ time\ when\ patient\ characteristic\ 1\ applies\ (minutes)$$

$n1 = $ *number of patients in the dayshift for whom characteristic* 1 *applies*

$b = $ *baseline care time per patient* (*minutes*)

$N = $ *number of patients admitted in the dayshift*

$x = $ *amount of care time for patient related activities* (*minutes*)

$z = $ % *of time spent by nurses on non* − *patient related activities*

**$Yreq = $ total required nursing time** (**minutes**)

Allocated nursing time (step 6) was determined by adding up the amount of nurses in the shift, multiplying this by the shift time and correcting for nurse proficiency.
The allocated nursing time was calculated as follows:

$$(m1 * p1 + m2 * p2 + m.. * p.. + ..) = Yall$$

$m1 = $ *amount of time nurse* 1 *was observed in the day shift* (*minutes*)

$p1 = $ *proficiency of nurse* 1

**$Yall = $ total allocated nursing time** (**minutes**)

The modelled workload (step 7) was calculated by dividing required nursing time by allocated nursing time. This ratio gives an objective indication of nurses' workload (W).
W = 100% reflects a perfect balance between allocated and required nursing staff; W<100% indicates overstaffing and W>100% understaffing.

$$W = 100\% * \frac{Yreq}{Yall}$$

**$Yreq = $ total required nursing time** (**minutes**)

**$Yall = $ total allocated nursing time** (**minutes**)

**$W = $ modelled workload** (%)

This modelled workload was calculated retrospectively for each ward on each of the day shifts during the time study period.

**Covariates: Personal resources, job resources and job demands.** Several covariates were included in the analysis, in order to determine interventions for balancing perceived workload in daily practice.

Personal resources were not expected to vary on a daily basis and were measured once at baseline, three weeks prior to the work sampling period. Personal resource proficiency was estimated by expert nurses, as mentioned above. Personal resource self-efficacy was measured by a scale based on Schwarzer & Jerusalem's validated general self-efficacy scale [59]. The scale contains five questions: 'When difficult problems occur at work, I know how to deal with

them', 'At work, I reach my goal, even when unexpected situations occur', 'If I encounter obstacles at work, I always find a way around them', 'Even if it takes a lot of my time and energy, at work I achieve what I want' and 'If I encounter something new at work, I always know how to deal with it'. All questions had a five point answer scale, ranging from 'Agree completely' (5) to 'Disagree completely' (1). Scores on the five questions were summed to a total score and then averaged per respondent.

Job resources support from management and support from colleagues were measured on a daily basis for all nurses on duty during the time study period. The questions originated from the validated Questionnaire on the Experience and Evaluation of Work (QEEW [30]). Support from management was measured by the question 'I could count on my supervisor when I came across difficulties in my work today'. Support from colleagues was determined by the question 'I could count on my colleagues when I came across difficulties in my work today'. Both questions had a 5 point answer scale, ranging from 'Not at all' (1) to 'Very often' (5).

The job resource percentage of registered nurses on the ward was determined by mapping the nurse qualifications of all nurses involved in the study and calculating a percentage of registered nurses on duty on each ward for each day of the time study.

The job demands perceived equality of work distribution and perceived interruptions were also measured once at baseline. Perceived equality of work distribution was measured by the question 'Is the work distributed evenly across all employees of the department?'. Perceived interruptions was determined by the question "Do you have to deal with interruptions in your work?". Both questions had a four point answer scale, ranging from 'Never' (1) to 'Always (4). These questions were derived from questionnaires often used by Dutch occupational health services.

The proportions of time spent on direct patient care and time spent on registration were derived from the observational study results [56] and determined per nurse per day. Direct patient care entails all activities that can be attributed to one specific patient, excluding registration. Registration is defined as all administrative and registration tasks done either on behalf of the patient (such as patient documentation, ordering medication and applying for tests and examinations) or as part of general administrative activities (such as ordering supplies, processing incoming mail).

## Data analysis

A one-way ANOVA and Tukey post-hoc tests were performed to determine significant differences between wards for the modelled workload.

Linear mixed effects models [60] were used to study the relation between the modelled workload and perceived workload. Observations were not independent since the majority of nurses were observed more than once during the work sampling period (longitudinal data). Linear mixed effects models are suitable for analyzing this type of data. The modelled workload $W$ was the independent variable and the five perceived workload measures were the outcome variables; each model included a random intercept per nurse to account for dependence of measures within nurses over time.

The relation between the modelled workload $W$ and the perceived workload was tested for each of the five perceived workload measures, with the wards as fixed effects. Subsequently, the personal resources (2 variables), job resources (3 variables) and job demands (4 variables) were added to the model in blocks to test the direct effects on perceived workload. In the last step, interactions between the modelled workload estimate and the four job demands and three job resources were introduced, to test for moderation. This resulted in five estimated models for each of the five outcome measures.

The significance level was set to 0.01. Model fits were evaluated for each outcome measure by comparing the Bayesian Information Criterion (BIC) values. BIC was chosen over the Akaike information criterion and likelihood ratio testing because the models include a relatively large number of independent variables and BIC is more conservative when testing several parameters at once.

### Ethical considerations

The study guaranteed the privacy of involved staff and patients. The study protocol was reviewed and approved by the medical ethical review board of the UMC Utrecht, protocol number 14-165/C.

## Results and discussion

### Baseline and daily questionnaires

The daily questionnaire was filled out 694 times, resulting in an average response rate of 58%. The baseline measure questionnaire was returned by 162 nurses; a response rate of 65%. Details on response rates, population characteristics and average responses for personal resources and job resources are shown per ward in **Table 1**.

Not surprisingly, the vast majority of nurses are female (**Table 1**). Of the respondents of the baseline questionnaire, between 44% and 70% were registered nurses; in the daily questionnaires this ranged between 37% and 68%. Gender and age category could not always be ascertained, for example for students who left the hospital or for temporary nursing staff. 36% to 68% of respondents were between 20 and 30 years old. The relatively high proportion of young nurses can be explained by the study setting: the research took place in an academic hospital, with specific training and educational tasks. On all wards, support of colleagues scored higher on average than support of management (4.0 to 4.3 and 3.7 to 4.1, respectively). Self-efficacy seems quite stable across wards, with average scores ranging between 3.5 and 3.7 and standard deviations between 0.4 and 0.5.

### Modelled workload

One of the main practical concerns in the study hospital was whether the workload was divided equally across wards. In order to answer this question, modelled workload was analyzed per ward. **Fig 2** presents the average modelled workload per day shift, calculated retrospectively for the days of the work sampling period. Each line represents one ward. During the work sampling period, on two day shifts equipment failed on a ward and on one day the data download failed for one ward. This meant that not all the required data were available to calculate the modelled workload on those days, so three day shifts were excluded for all wards. Days 1, 3 and 5 are missing. This gives a sample size of 12 shifts x 6 wards = 72 observations of modelled workload on ward level. Since these failures occurred randomly, it is expected that the missing data do not influence the outcomes.

The post hoc tests found that modelled workload on ward 6 was significantly lower than on wards 1 (estimated difference -32.3%, confidence interval -58.1% to -6.4%, p-value 0,001) and 4 (estimated difference -36.6%, confidence interval -62.4% to -10.7%, p-value <0,001), and workload on ward 3 was significantly lower than on ward 4 (estimated difference 30.2%, confidence interval -56.0% to -4.3%, p-value 0,002).

### Perceived workload

For the perceived workload measures, data were available for all fifteen day shifts in the work sampling period.

**Table 1. Response rates, population characteristics and average response of the respondents of the baseline and daily questionnaires.**

| Ward | | 1 | | | | 2 | | | | 3 | | | | 4 | | | | 5 | | | | 6 | | |
|---|---|---|---|---|---|---|---|---|---|---|---|---|---|---|---|---|---|---|---|---|---|---|---|---|
| Response | | n | % | Mean | SD | n | % | | | n | % | | | n | % | | | n | % | | | n | % | | |
| Baseline response | | 28 | 72% | | | 33 | 73% | | | 29 | 54% | | | 31 | 62% | | | 16 | 53% | | | 25 | 83% | | |
| Gender | Male | 2 | 7% | | | 3 | 9% | | | 5 | 17% | | | 6 | 19% | | | 0 | 0% | | | 2 | 8% | | |
| | Female | 26 | 93% | | | 30 | 91% | | | 24 | 83% | | | 25 | 81% | | | 16 | 100% | | | 23 | 92% | | |
| Age category | <20 | 1 | 4% | | | 0 | 0% | | | 0 | 0% | | | 0 | 0% | | | 0 | 0% | | | 1 | 4% | | |
| | 20-30 | 10 | 36% | | | 15 | 45% | | | 17 | 59% | | | 21 | 68% | | | 8 | 50% | | | 13 | 52% | | |
| | 30-40 | 3 | 11% | | | 9 | 27% | | | 4 | 14% | | | 3 | 10% | | | 2 | 13% | | | 3 | 12% | | |
| | 40-50 | 3 | 11% | | | 4 | 12% | | | 1 | 3% | | | 1 | 3% | | | 0 | 0% | | | 2 | 8% | | |
| | >=50 | 8 | 29% | | | 2 | 6% | | | 3 | 10% | | | 2 | 6% | | | 5 | 31% | | | 5 | 20% | | |
| | Unknown | 3 | 11% | | | 3 | 9% | | | 4 | 14% | | | 4 | 13% | | | 1 | 6% | | | 1 | 4% | | |
| Type of nurse | Registered nurse | 24 | 86% | | | 31 | 94% | | | 20 | 69% | | | 21 | 68% | | | 10 | 63% | | | 16 | 64% | | |
| | Student nurse | 4 | 14% | | | 2 | 6% | | | 9 | 31% | | | 10 | 32% | | | 6 | 37% | | | 9 | 36% | | |
| Self-efficacy (range 1-5) | | | | 3.5 | 0.4 | | | 3.6 | 0.5 | | | 3.6 | 0.4 | | | 3.7 | 0.4 | | | 3.6 | 0.4 | | | 3.6 | 0.5 |
| Interruptions (range 1-4) | | | | 2.5 | 0.8 | | | 2.7 | 0.7 | | | 2.3 | 0.7 | | | 2.3 | 0.8 | | | 2.4 | 0.8 | | | 2.0 | 0.9 |
| Equal work distribution (range 1-4) | | | | 2.8 | 0.5 | | | 2.5 | 0.7 | | | 2.7 | 0.6 | | | 2.6 | 0.7 | | | 2.8 | 0.4 | | | 2.6 | 0.7 |
| Daily response | | 91 | 58% | | | 107 | 45% | | | 141 | 63% | | | 151 | 66% | | | 93 | 57% | | | 111 | 61% | | |
| Gender | Male | 15 | 16% | | | 3 | 3% | | | 36 | 26% | | | 30 | 20% | | | 8 | 9% | | | 12 | 11% | | |
| | Female | 68 | 75% | | | 96 | 90% | | | 102 | 72% | | | 95 | 63% | | | 80 | 86% | | | 91 | 82% | | |
| | Unknown | 8 | 9% | | | 8 | 7% | | | 3 | 2% | | | 26 | 17% | | | 5 | 5% | | | 8 | 7% | | |
| Age category | <20 | 3 | 3% | | | 0 | 0% | | | 0 | 0% | | | 11 | 7% | | | 4 | 4% | | | 12 | 11% | | |
| | 20-30 | 49 | 54% | | | 66 | 62% | | | 106 | 75% | | | 98 | 65% | | | 62 | 67% | | | 67 | 60% | | |
| | 30-40 | 8 | 9% | | | 16 | 15% | | | 12 | 9% | | | 9 | 6% | | | 19 | 20% | | | 18 | 16% | | |
| | 40-50 | 13 | 14% | | | 15 | 14% | | | 8 | 6% | | | 4 | 3% | | | 0 | 0% | | | 3 | 3% | | |
| | >=50 | 10 | 11% | | | 2 | 2% | | | 12 | 9% | | | 3 | 2% | | | 3 | 3% | | | 3 | 3% | | |
| | Unknown | 8 | 9% | | | 8 | 7% | | | 3 | 2% | | | 26 | 17% | | | 5 | 5% | | | 8 | 7% | | |
| Type of nurse | Registered nurse | 62 | 68% | | | 70 | 65% | | | 68 | 48% | | | 85 | 56% | | | 51 | 55% | | | 41 | 37% | | |
| | Student nurse | 29 | 32% | | | 37 | 35% | | | 73 | 52% | | | 66 | 44% | | | 42 | 45% | | | 70 | 63% | | |
| Support management (range 1-5) | | | | 3.8 | 1.1 | | | 3.7 | 1.2 | | | 4.1 | 1.1 | | | 3.9 | 1.0 | | | 3.9 | 1.1 | | | 4.1 | 1.0 |
| Support colleagues (range 1-5) | | | | 4.1 | 0.8 | | | 4.0 | 0.9 | | | 4.2 | 0.9 | | | 4.0 | 0.9 | | | 4.2 | 0.8 | | | 4.3 | 0.8 |

Fig 3 presents the average perceived workload per ward per day shift, for each of the five perceived workload measures.

The graphs of amount of work and work pace are quite similar. The graph of emotional load is more stable and on average lower than the other measures. To a lesser extent, the same goes for the results on physical load. On day 4 on ward 2, workload was apparently perceived exceptionally high on all of the five outcomes measures. On day 9 on ward 2, all workloads were perceived to be relatively low, but mental load peaked.

## Effects of personal resources, job resources and job demands

In total, the study includes 351 observations where both the objective and subjective workload measures were available for a nurse on a particular day. For all five perceived workload measures, the models with interactions did not perform better than the models without interactions, hence the interactions are not shown in Table 2. This means that no moderation was found on the relation between modelled and perceived workload, all significant effects of variables were direct effects on perceived workload.'

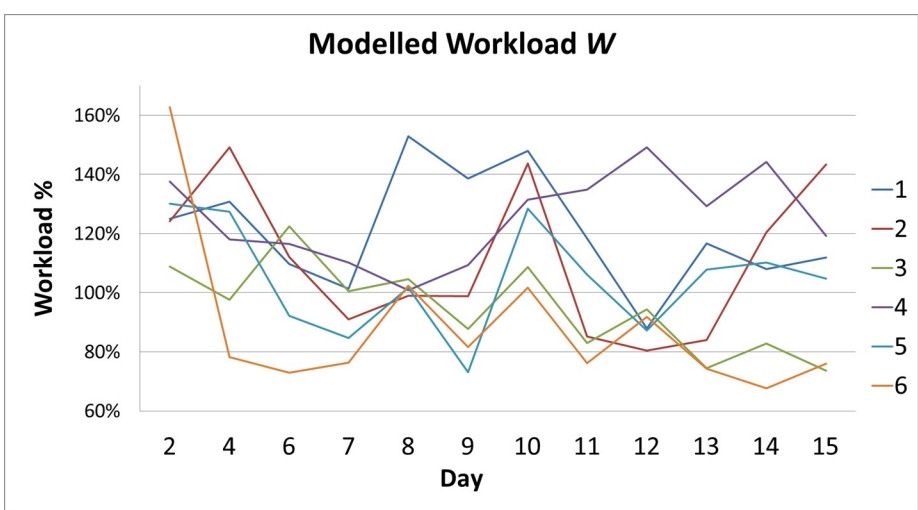

**Fig 2. Average modelled workload per ward per day, calculated retrospectively for the work sampling period.**

For three outcome measures (mental load, emotional load and physical load), the models that included the job demands did not perform better than the models without the job demands. Therefore, no results are shown for the job demands for these three outcome measures.

The results indicate a positive correlation between all perceived workload measures and modelled workload. Every 10% increase in estimated workload (on a scale that ranges roughly between 70 and 170%) is associated with a 0.209 increase in perceived work pace (on a scale from 1 to 5 this is a 5.2% increase), 0.198 (5%) in perceived amount of work, 0.141 (3.5%) in perceived mental workload, 0.043 (1.1%) in perceived emotional workload and 0.047 (1.2%) in perceived physical load.

Neither of the personal resources were significantly related to any of the perceived workload measures. The scale for self-efficacy was tested for internal consistency by calculating

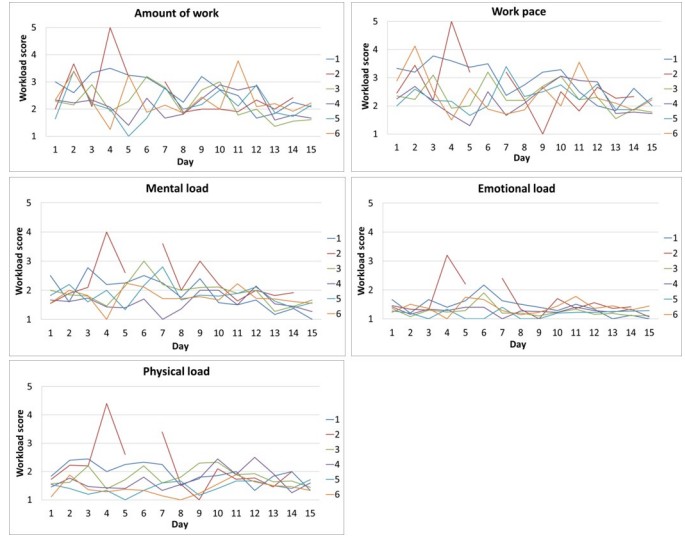

**Fig 3. Five line graphs with average perceived workloads per ward per day for amount of work, work pace, mental load, emotional load and physical load.**

**Table 2. Effects of job demands, job resources and personal resources on five outcome measures of perceived workload (range 1 to 5).**

| Type of variable | Variable | Perceived Work pace | | Perceived Amount of work | | Perceived Mental load | | Perceived Emotional load | | Perceived Physical load | |
|---|---|---|---|---|---|---|---|---|---|---|---|
| | | Estimated effect [a] | Standard error | Estimated effect [a] | Standard error | Estimated effect [a] | Standard error | Estimated effect [a] | Standard error | Estimated effect [a] | Standard error |
| **Job demand** | Modelled workload | **0.209** | **0.024** | **0.198** | **0.025** | **0.141** | **0.021** | **0.043** | **0.013** | **0.047** | **0.017** |
| **Personal resource** | Perceived self-efficacy | 0.010 | 0.211 | -0.03 | 0.248 | -0.12 | 0.212 | -0.08 | 0.147 | -0.10 | 0.218 |
| | Estimated nurse proficiency % | -0.00 | 0.039 | -0.03 | 0.046 | 0.008 | 0.037 | -0.00 | 0.026 | -0.00 | 0.038 |
| **Job resource** | Perceived support of management | -0.01 | 0.057 | -0.05 | 0.061 | -0.08 | 0.052 | -0.02 | 0.032 | -0.03 | 0.043 |
| | Perceived support of colleagues | **-0.23** | **0.073** | **-0.30** | **0.079** | **-0.23** | **0.067** | **-0.11** | **0.042** | **-0.22** | **0.057** |
| | % of nurses on ward that is registered nurse | 0.088 | 0.057 | 0.068 | 0.061 | 0.083 | 0.052 | **0.085** | **0.032** | 0.076 | 0.044 |
| **Job demand** | % nursing time spent on direct patient care | **0.165** | **0.037** | **0.228** | **0.040** | . | . | . | . | . | . |
| | % nursing time spent on registration | -0.19 | 0.080 | **-0.29** | **0.085** | . | . | . | . | . | . |
| | Perceived equality in work distribution | -0.10 | 0.171 | -0.15 | 0.201 | . | . | . | . | . | . |
| | Perceived interruptions | 0.232 | 0.120 | 0.150 | 0.142 | . | . | . | . | . | . |

[a] For variables with ordinal scales (for example from "Never" to "Always", translated into 1 to 4 points), a rise of one full point results in the effect shown in **Table 3**. For ratio variables, a rise of 10% results in the effect shown in this table. Significant effects are printed in bold.

Cronbach's alpha. Alpha was 0.657 and did not improve if items were deleted from the scale. Since this practical scale is used often by occupational health services, it was decided to keep the scale in the study and test whether it was of influence on perceived workload even though Cronbach's alpha was relatively low.

Support from management was not related to any of the perceived workload measures, whereas support from colleagues was negatively associated with all outcome measures. A 1-point increase (on an scale of 1 to 5) in experienced support from colleagues reduced the experienced work pace with 0.23 (5.8%), amount of work with 0.30 (7.5%), mental workload with 0.23 (5.8%), emotional workload with 0.11 (2.8%) and physical load with 0.22 (5.5%).

The percentage of registered nurses on the ward is correlated with the perceived emotional workload, but not the other subjective workload outcomes. Every 10% increase in proportion of registered nurses on a ward results in 0.085 (2.1%) increase in experienced emotional workload.

The proportion of time spent on direct patient care and the proportion of time spent on registration were significantly related to perceived workload. For each 10% increase in time spent on direct patient care, there was a 0.165 (4.1%) point rise in perceived work pace and 0.228 (5.7%) point rise in perceived amount of work. Every 10% increase in proportion of time spent on registration gave a 0.29 (7.3%) decline in perceived amount of work. The other job demands were not significantly related to any of perceived workload measures.

**Table 3. Proportional effects for variables with significant effect on perceived workload.**

| Type of variable | Variable | Increase | Perceived Work pace | Perceived Amount of work | Perceived Mental load | Perceived Emotional load | Perceived Physical load |
|---|---|---|---|---|---|---|---|
| | | | Estimated proportional effect* | Estimated proportional effect* | Estimated proportional effect* | Estimated proportional effect* | Estimated proportional effect* |
| Job demand | Modelled workload | +10% | +5.2% | +5% | +3.5% | +1.1% | +1.2% |
| Job resource | Perceived support of colleagues | +1 point | -5.8% | -7.5% | -5.8% | -2.8% | -5.5% |
| | % of nurses on ward that is registered nurse | +10% | | | | +2.1% | |
| Job demand | % nursing time spent on direct patient care | +10% | +4.1% | +5.7% | | | |
| | % nursing time spent on registration | +10% | | -7.3% | | | |

Estimated effect of a rise of either 10% or 1 point in the independent variable on the outcome variables.

Proportional effects for significant variables are summarized in Table 3.

## Discussion

### Findings

The modelled workload is significantly different between wards 6 and 1, between 6 and 4 and between 3 and 4. The estimated differences are quite large; -32.3%, -36.6% and -30.2% respectively but so are the corresponding confidence intervals: -58.1% to -6.4%, -62.4% to -10.7% and -56.0% to -4.3% respectively.

The *first hypothesis* was accepted, since there is a linear correlation between the modelled workload and all five perceived outcome measures. Every 10% increase in modelled workload results in a 0.209 (5.2%) increase in perceived work pace, 0.198 (5%) in perceived amount of work, 0.141 (3.5%) in perceived mental workload but only 0.043 (1.1%) in perceived emotional workload and 0.047 (1.2%) in perceived physical load. Apparently, modelled workload *W* has the biggest effect on work pace, amount of work and mental workload. This makes sense, considering that the modelled workload *W* is based on comparing required and allocated care time only, and does not give any insight in emotional and physical requirements. Having to work fast due to a lack of balance between required and allocated resources may result in an experienced higher mental load because tasks that need focus and concentration need to be done under time pressure. Modelled workload, as found in this study, varies roughly between 70% and 170%; a range of 100%. With every 10% increase of the modelled workload, the perceived work pace increases with 5.2% and the perceived amount of work with 5.0%. It seems that perceived workload does not rise to the same extent as modelled workload. However, we need to consider that the answer range that was provided for all perceived workload measures ranges from 1 to 5 on an ordinal scale, which is quite a narrow range. Respondents therefore had limited options to express their perceptions of workload. In future research, a broader range of response may be of value to get a more detailed insight on the extent to which modelled workload influences perceptions.

Of the graphs that concern perceived workload, work pace and amount of work show a similar pattern. In the QEEW [30] these two are studied under one construct: work pressure. In our study, we studied them separately. We found that the percentage of time spent on registration was correlated with perceived amount of work but not with work pace. Apparently, researching these two measures as separate constructs pays off.

The graph of emotional load is most stable over time and reports a lower average than the other measures. To a lesser extent, the same goes for physical load.

On ward 2, workload peaked on days 4 and 9, we could not explain why. Especially the peak in mental workload on day 9 is puzzling, since all other workloads measures on that day were relatively low.

Personal resources self-efficacy and proficiency were not correlated with any of the outcome measures, so *hypothesis two* was rejected. As for self-efficacy, this is not in line with findings in other studies, such as those of Spence Laschinger [51] and Martens [53], where positive correlations were found between (occupational coping) self-efficacy and workload and stress of staff shortages. Our findings do correspond with Schmidt's [41], who found an effect of self-efficacy on job strain. After additional analysis, the internal consistency of the scales for self-efficacy used in our study turned out to be low, so no definite conclusion can be drawn about the effect of self-efficacy on perceived workload. It is recommended that in further research, other validated scales for self-efficacy are used.

Personal resource proficiency of nurses was not significantly related to workload in our study, which is. in line with a study on nurse risk assessment decisions [61]. This study showed that under conditions without time pressure, nurses with clinical expertise (i.e. more proficient nurses) performed better than novice nurses. However, the positive effects of clinical expertise were negated when time pressure was introduced to clinical simulations.

For all outcome measures, the models without the interactions showed a better fit than the models with interactions. This means that there is only evidence for direct effects, there is no moderation, which means that *hypotheses three and four* must be rejected.

We did however find direct effects. The job resources support from colleagues and proportion of registered nurses were significantly related to perceived workload.

Support from colleagues proved to be an important factor in workload perception, since it was negatively correlated to all outcome measures. This is in line with findings of other studies [7, 44, 45], where teamwork was shown to offset negative effects of high workload. The fact that the effect of teamwork was found to be less strong on outcome measure perceived emotional workload was unexpected. Apparently support from colleagues is more important in handling workload in the cognitive and physical domain than when coping with emotional challenges. The percentage of registered nurses on the ward is positively correlated with the perceived emotional workload, although not very strong. It was expected that registered nurses would be better equipped to handle emotional stressors than student nurses, but apparently, there is more to this. Possibly, registered nurses feel more responsible for their work than student nurses, which may increase emotional load. Or maybe registered nurses are more likely to be given the more emotionally challenging tasks than student nurses. More research is needed to explain this effect. Perceived support from management did not turn out to be significantly related to any of the outcome measures. A study by MacPhee [14] did show that support of unit-level management in managing workload could have positive effects on perceived quality of care and job outcomes, but this study did not test the direct relation between support of management and perceived workload. Apparently sticking together as a team is more important in workload perceptions than experiencing direct support from ward leadership. This suggests that when workload is high, teams may benefit from efforts to improve team cohesion. Ward leadership has an important role in enhancing teamwork and teambuilding on the ward.

Job demands proportion of direct patient care and proportion of registration were significantly related to perceived workload. An increased proportion of direct patient care results in increased perceptions of work pace and amount of work. This may be explained by the fact that time spent with the patient is regarded as more demanding; possibly patient related tasks

are considered more urgent than other tasks and add to perceived work pressures. In line with this finding, an increase in proportion of time spent on registration gave a decline in perceived amount of work. Registration might be regarded as a less demanding task which may influence the perception of the work pressure. More research is needed on this, but the results may be explained by the fact that registration takes place in a quiet environment, away from the patient and thus away from potential pressures of patients and relatives or other caregivers. Possibly administrative work can form a stable, quiet moment in the working day where nurses can focus on this one task, whilst sitting down at a desk, instead of having to hurry along between tasks in different patient rooms. It is also possible that registration work is saved up for quieter days, when there is time to catch up on registration.

The job demand considering the level of perceived work interruptions seemed irrelevant to perceived workload. Literature [35, 37] shows a negative correlation between interruptions and patient safety, for example when performing complicated tasks such as medication preparation. Therefore a positive correlation between interruptions and perceived workload was expected, at least with outcome measure perceived mental load.

Perceived equality in work distribution was also not correlated to any of the outcome measures. To our knowledge there is no available literature on the effects of equal work distribution on perceived workload. There are studies on distributive organizational justice, which reflects the perceived fairness in decision outcomes, such as work scheduling). These suggest a negative correlation between distributive justice and intention to leave [62] and a positive correlation between distributive justice and quality of work life (for example psychological well-being, workload and work satisfaction) [63]. We assumed that equal work distribution would be interpreted as fair distribution and thus a positive correlation between equal work distribution and perceived workload was expected This may be due to the fact that this job demand was only measured once instead of daily.

More research is recommended on the effects of perceived interruptions and perceived equality of work distribution.

## Strengths and limitations

This was a longitudinal, multilevel study, which is rare in this field of research. The fact that an objective (modelled) and subjective (perceived) workload measures were combined in one study is unique.

The effects of the patient characteristics on care time were quantified by work sampling over a relatively long study period of 15 day shifts.

Nurse proficiency was included in the workload calculations, which has not been done before and proved to be an important factor to include in the analysis. Correcting for nurse proficiency is especially relevant in a teaching hospital setting, where proficiency of nurses varies greatly.

The combination of a modelled workload measure and five perceived workload measures in one study is unique. Also, testing for moderation of job demands and job resources on the relation between the modelled and perceived workloads has not been done before.

The overall response rates on the baseline measure questionnaire and the daily questionnaire that was used during the work sampling period were 65% and 70% respectively, which is considered to be quite good [64].

There are several limitations that need to be addressed. This study was set in an academic hospital, which makes it uncertain whether the study results can be readily applied to different settings, such as general hospitals. Nurses' activities and the patient mix in general hospitals are likely to be different than in academic hospitals, if this study is to be applied in such a

setting, the framework can be the same, but it is recommended to review the list of patient characteristics and to repeat the work sampling. The same goes for departments with a unique nature, such as pediatric and psychiatric wards.

The confidence intervals for the estimated differences of modelled workload between wards were relatively high. For future research, a larger sample size is recommended in order to make a more accurate estimate of differences between wards.

The scales used to measure self-efficacy were chosen for practical reasons and were not validated by means of international publications. The items on the scale proved not to be internally consistent; future research should include validated scales of self-efficacy. Since the items do not represent one construct, no conclusion can be drawn about the effect of self-efficacy on perceived workload.

Perceived equality of work distribution and perceived interruptions were both measured by one question. Possibly, the use of more extensive measures would have influenced results. More research is recommended on this.

In this study, proficiency was estimated by a mini-Delphi study among head nurses. Another way to measure the proficiency of nurses would be to keep track of the actual exact time spent on each activity, calculate an estimate per activity per type of nurse and derive the proficiency percentage from these estimates. However, since there were 6 types of nurses and 24 activity groups in the study, this approach would have required a much larger sample size and a more accurate measurement of time spent on an activity than work sampling every ten minutes. For practical reasons (costs, registration), this was not possible, and the choice was made to estimate nurse proficiency instead.

Lastly, the work sampling study was done during fifteen consecutive dayshifts, which is quite a long time period but also only one time period. Possibly, results would have been more powerful if the study was done over two separate time periods. However, due to practical issues we did not chose this option. If the study was set over two time periods, staff would need to be trained and informed twice (for example not all medical students would be available for both time periods and new nursing students would have entered the wards), equipment would need to be rented twice and the preparations such as barcoding patient rooms would need to be done again. This would have become too costly so the study team opted against this.

## Conclusions

This paper presents a method to determine an objective measure for workload of nurses. This measure is positively associated with the perceived workload and can be used to detect differences in workload between wards. This may help in distributing workload more evenly, in order prevent over- and understaffing and issues in the domain of e.g. organizational justice. Job resources 'support of colleagues' and job demands 'time spent on direct patient care' and 'time spent on registration' had the biggest significant effects on perceived workload. When workload is high, extra effort in teambuilding is likely to have a positive effect on perceived workload. Unit-level management can contribute to reducing perceived workload by facilitating the nursing team to work together smoothly and by enhancing team spirit. Since time spent on direct patient care is positively associated with perceived workload and time spent on registration is negatively associated with perceived workload, a good balance between time spent on direct patient care and registration may also benefit perceptions of workload. Also, registered nurses experience a higher workload when it comes to the amount of work than other nurses. Further research is recommended.

The findings of this research can help nurse management in allocating resources and directing their attention to the most relevant factors so workload of nurses is better balanced, which

in turn leads to a higher quality of care, keeping nurses healthy and the prevention of additional costs for overstaffing, absenteeism or high turnover of nurses.

## Author Contributions

**Conceptualization:** Wilhelmina F. J. M. van den Oetelaar, Corné A. M. Roelen, Wilko Grolman, Rebecca K. Stellato, Willem van Rhenen.

**Data curation:** Wilhelmina F. J. M. van den Oetelaar.

**Formal analysis:** Wilhelmina F. J. M. van den Oetelaar, Rebecca K. Stellato.

**Investigation:** Wilhelmina F. J. M. van den Oetelaar.

**Methodology:** Wilhelmina F. J. M. van den Oetelaar, Corné A. M. Roelen, Wilko Grolman, Rebecca K. Stellato, Willem van Rhenen.

**Project administration:** Wilhelmina F. J. M. van den Oetelaar.

**Supervision:** Corné A. M. Roelen, Willem van Rhenen.

**Validation:** Wilhelmina F. J. M. van den Oetelaar, Corné A. M. Roelen, Rebecca K. Stellato.

**Visualization:** Wilhelmina F. J. M. van den Oetelaar.

**Writing – original draft:** Wilhelmina F. J. M. van den Oetelaar.

**Writing – review & editing:** Wilhelmina F. J. M. van den Oetelaar, Corné A. M. Roelen, Wilko Grolman, Rebecca K. Stellato, Willem van Rhenen.

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
