## [Decision Letter · Decision Letter 0]

12 Nov 2020

PONE-D-20-08785

Exploring the relation between modelled workload and perceived workload of nurses and related job demands, job resources and personal resources, a longitudinal study

PLOS ONE

Dear Dr. van den Oetelaar,

Thank you for submitting your manuscript to PLOS ONE. After careful consideration, we feel that it has merit but does not fully meet PLOS ONE’s publication criteria as it currently stands. Therefore, we invite you to submit a revised version of the manuscript that addresses the points raised during the review process.

We look forward to receiving your revised manuscript.

Kind regards,

Yong-Hong Kuo

Academic Editor

PLOS ONE

Additional Editor Comments (if provided):

Based on the reviewers' recommendations and comments, I recommend major revisions. Please seriously address the reviewers' concerns.

Journal Requirements:

2) Please include captions for your Supporting Information files at the end of your manuscript, and update any in-text citations to match accordingly. Please see our Supporting Information guidelines for more information: http://journals.plos.org/plosone/s/supporting-information.

3) Please ensure that you refer to Figure 1 in your text as, if accepted, production will need this reference to link the reader to the figure.

4) Thank you for stating the following financial disclosure:

 [NO. The funders had no role in study design, data collection and analysis, decision to

publish, or preparation of the manuscript].

Reviewers' comments:

Reviewer's Responses to Questions

**Comments to the Author**

1. Is the manuscript technically sound, and do the data support the conclusions?

Reviewer #1: Partly

Reviewer #2: Yes

2. Has the statistical analysis been performed appropriately and rigorously? 

Reviewer #1: No

Reviewer #2: Yes

3. Have the authors made all data underlying the findings in their manuscript fully available?

Reviewer #1: Yes

Reviewer #2: Yes

4. Is the manuscript presented in an intelligible fashion and written in standard English?

Reviewer #1: Yes

Reviewer #2: No

5. Review Comments to the Author

Reviewer #1: The paper is very difficult to follow as it does not follow normal model development and presentation and the format is really not well done. There are so many tables and finally it is difficult to follow what was found. There was no hypotheses development and the there were no directions in the hypotheses stated. The moderation hypotheses are also not well developed and stated. Why study only one academic hospital, as they have very different set up from normal hospital and why this particular hospital. The discussions and implications are also difficult to follow.

Reviewer #2: General comments and recommendations

1.Sample size and sample size estimation methods was not stated

2.The sampling technique information was not described

3.Data collection tools was not stated to its development i.e if self prepared, standard, its validity and reliability.

4.How did the tools delivered to the study participants? i.e did it via self-administrated, interview or any other....?

5.The ways of language sentences formulation needs revision specially "tenses"

6.The title, the objective, discussion, and conclusion strictly needs to be in line.

6. PLOS authors have the option to publish the peer review history of their article (what does this mean?). If published, this will include your full peer review and any attached files.

Reviewer #1: No

Reviewer #2: No

---

## [Author Response · Author response to Decision Letter 0]

21 Dec 2020

Enclosed is a revision of a manuscript to be considered for publication in PLOS ONE. The manuscript was first submitted to you on December 14th 2019. A first review round has been returned to us on the 17th of February 2020. On March 30th a revised manuscript was submitted. This revised manuscript has been reviewed and reviewer suggestions have been sent to us on November 12th 2020. The reviewers suggest major revisions. We have carefully looked at the reviewers’ suggestions and would like to respond in this letter. We have addressed all points put forward by the referees and believe the manuscript has significantly gained in clarity. Corrections have been made in the main manuscript, using ‘Track changes’ functionality. The new versions of the main document, with and without visible changes, have been re-submitted through your website. The following revisions have been made. Please note that the references to lines in the manuscript refer to the lines in the final document without tracked changes. 

Reviewer 1:

1. The paper is very difficult to follow as it does not follow normal model development and presentation and the format is really not well done. There are so many tables and finally it is difficult to follow what was found. There was no hypotheses development and the there were no directions in the hypotheses stated. The moderation hypotheses are also not well developed and stated. Why study only one academic hospital, as they have very different set up from normal hospital and why this particular hospital. The discussions and implications are also difficult to follow.

• The introduction, method and discussion sections have been thoroughly revised in order to clarify the purpose of the research, development of the hypotheses and overall ease of reading

• Clarification on directions has been added to the hypotheses, see lines 189-197

• Reasoning behind moderation is explained in line 199-201

• The amount of tables in the manuscript is reduced to 3

• The study originated from the urgent practical need for a workload management tool in this particular academic hospital. At the time it was decided to first attempt to develop a practical tool to use in this hospital and if successful, subsequently validate this tool in other academic and regional hospitals. 

Reviewer 2: 

1. Sample size and sample size estimation methods was not stated.

Sample sizes of all questionnaires in the study can be found in Table 1. Sample size of the modelled workload on ward level was included in the results section, see line 417. Sample size of observations for the model that studies the relation between the objective and subjective workload of nurses is also added to the results section, see line 444.

2. The sampling technique information was not described 

Data collection method was described in the method section, see line 230-248 

3. Data collection tools was not stated to its development i.e if self prepared, standard, its validity and reliability 

Validity of all questionnaires used in this study is discussed throughout the method section see lines 335 and 345 and also in the discussion section, see line 539, 631.

4. How did the tools delivered to the study participants? i.e did it via self-administrated, interview or any other....?.

The daily questionnaire was handed out on paper at the end of the shift by the lead researcher, the baseline questionnaire was sent via Survey Monkey. This information is added to the method section, see line 247. 

5. The ways of language sentences formulation needs revision specially "tenses"

A native English speaker has revised the manuscript, corrections have been made where needed

6. The title, the objective, discussion, and conclusion strictly needs to be in line).” 

Revisions have been made throughout the introduction, method and discussion sections to align this

7. The study is considered longitudinal because study objects are observed multiple times, although it is not longitudinal in the classical sense, where the exact same group is followed on specific points in time. The study is considered to be multi-level because data are analyzed per ward and per nurse. 

1. Style requirements and file names were checked and adjusted where necessary

2. There is no longer supporting information in the manuscript

3. References to all figures and tables are checked with journal requirements

4. The authors received no specific funding for this work

We sincerely thank you for all your help and suggestions so far, and look forward to your response. 

Kind regards, 

W.F.J.M. (Miranda) van den Oetelaar, PhD, MSc.

---

## [Decision Letter · Decision Letter 1]

25 Jan 2021

Exploring the relation between modelled and perceived workload of nurses and related job demands, job resources and personal resources: a longitudinal study

PONE-D-20-08785R1

Dear Dr. van den Oetelaar,

We’re pleased to inform you that your manuscript has been judged scientifically suitable for publication and will be formally accepted for publication once it meets all outstanding technical requirements.

Kind regards,

Yong-Hong Kuo

Academic Editor

PLOS ONE

Additional Editor Comments (optional):

Based on the reviewer's recommendation, I recommend Accept.

Reviewers' comments:

Reviewer's Responses to Questions

**Comments to the Author**

1. If the authors have adequately addressed your comments raised in a previous round of review and you feel that this manuscript is now acceptable for publication, you may indicate that here to bypass the “Comments to the Author” section, enter your conflict of interest statement in the “Confidential to Editor” section, and submit your "Accept" recommendation.

Reviewer #1: All comments have been addressed

2. Is the manuscript technically sound, and do the data support the conclusions?

Reviewer #1: Yes

3. Has the statistical analysis been performed appropriately and rigorously? 

Reviewer #1: Yes

4. Have the authors made all data underlying the findings in their manuscript fully available?

Reviewer #1: Yes

5. Is the manuscript presented in an intelligible fashion and written in standard English?

Reviewer #1: Yes

6. Review Comments to the Author

Reviewer #1: The authors have revised and provided some justification although they can be contested but the revisions are acceptable.

7. PLOS authors have the option to publish the peer review history of their article (what does this mean?). If published, this will include your full peer review and any attached files.

Reviewer #1: No

---

## [Editor Report · Acceptance letter]

17 Feb 2021

PONE-D-20-08785R1 

Exploring the relation between modelled and perceived workload of nurses and related job demands, job resources and personal resources; a longitudinal study. 

Dear Dr. van den Oetelaar:

I'm pleased to inform you that your manuscript has been deemed suitable for publication in PLOS ONE. Congratulations! Your manuscript is now with our production department. 

Kind regards, 

on behalf of

Dr. Yong-Hong Kuo 

Academic Editor

PLOS ONE